# Understanding the Effects of Growing Seasons, Genotypes, and Their Interactions on the Anthesis Date of Wheat Sown in North China

**DOI:** 10.3390/biology10100955

**Published:** 2021-09-24

**Authors:** Ziwei Li, Bangyou Zheng, Yong He

**Affiliations:** 1Institute of Environment and Sustainable Development in Agriculture, Chinese Academy of Agricultural Sciences, Beijing 100081, China; ziweiq866@126.com; 2CSIRO Agriculture and Food, Queensland Bioscience Precinct, 306 Carmody Road, St. Lucia, QLD 4067, Australia; bangyou.zheng@csiro.au

**Keywords:** *Triticum aestivum* L., anthesis date, additive main effects and multiplicative interaction model, growing season–genotype interactions

## Abstract

**Simple Summary:**

Wheat anthesis date is an important turning point for wheat from vegetative growth stage to reproductive growth stage, which is crucial for wheat to adapt to environment and increase grain yield. In this study, a panel of adaptive wheat varieties including historical varieties from the 1940s and current varieties was used to understand the contribution of growing season, genotypes and their interaction effects to anthesis date. Based on our results, we can conclude that growing seasons contributed tremendously to the anthesis date of wheat. In future wheat breeding, more consideration should be given to growing seasons, and the gene combination with the strongest adaptability to the growing seasons should be selected.

**Abstract:**

Quantitative studies on the effects of growing season, genotype (including photoperiod genes and vernalization genes), and their interaction (GGI) on the anthesis date of winter wheat (*Triticum aestivum* L.) are helpful to provide a scientific reference for selecting or developing adaptive varieties in target environments. In this study, we collected 100 winter wheat varieties with ecological adaptability in North China and identified the anthesis date under field conditions for three consecutive years from 2016 to 2019 with mapped photoperiod and vernalization alleles. Our results showed that the number of the photoperiod-insensitive *Ppd-D1a* allele increased with variety replacement, while the haplotype *Ppd-A1b* + *Ppd-D1b* + *vrn-D1* (A4B2) decreased from the 1940s to 2000s. The anthesis date of A4B2 was significantly delayed due to the photoperiod-insensitive alleles *Ppd-A1b* and *Ppd-D1b*. The additive main effect and multiplicative interaction (AMMI) model and GGI biplot analysis were used for data analysis. A large portion of the total variation was explained by growing seasons (66.3%), while genotypes and GGIs explained 21.9% and 10.1% of the anthesis dates, respectively. The varieties from the 1940s and before had a great influence on the anthesis date, suggesting these germplasms tend to avoid premature anthesis and could facilitate the development of phenological resilient varieties.

## 1. Introduction

Anthesis is a critical turning point in the transition from the vegetative to the reproductive growth stage of wheat (*Triticum aestivum* L.) [1] and is important for improving the adaptability of wheat to abiotic (temperature and water) and biotic stress (pest and disease) as well as for increasing the seed set [2,3]. Therefore, the anthesis date is a key period in the entire phenology of wheat and one of the key determinants of wheat yield [4]. Many studies have shown that temperature and photoperiod are the main factors affecting the anthesis date of wheat under sufficient light and irrigation [5,6,7]. The growing seasonal temperature fluctuation does not only affect photosynthesis, nutrient uptake, and respiration of wheat [8] but also anthesis during its normal phenological growth [9].

Crop breeding is a scientific direction based on adaptation of cropping systems to local climate. Varieties with early anthesis can adapt to a shorter growing season, while varieties with late anthesis are conducive to storing resources throughout the longer growing season without water deficit stress [10]. Farmers and breeders can make wheat production more flexible and adapt it to different geographical locations by selecting wheat varieties with different anthesis onsets to improve wheat yield [10]. Therefore, the research on the influence of growing seasons on anthesis date and the change in wheat anthesis onset can provide a scientific basis for breeding wheat varieties that can adapt to specific climate conditions.

Genetic diversity is a reason for the difference in anthesis date in wheat [11], and gene polymorphism analysis is the most direct and effective method in genetic diversity research [12]. Photoperiod and vernalization alleles play an important role in anthesis date by regulating the sensitivity of wheat to photoperiod and vernalization, respectively [13]. Photoperiod sensitivity is mainly controlled by three homologous genes of *Ppd-1*, namely, *Ppd-A1*, *Ppd-B1*, and *Ppd-D1* [14]. Previous studies have shown that the dominant allele in *Ppd-1* causes photoperiod insensitivity to wheat, while the photoperiod insensitivity caused by *Ppd-D1a* is greater than that caused by both *Ppd-B1a* and *Ppd-A1a* [15]. In terms of the relationship between photoperiod sensitivity and anthesis, photoperiod insensitivity alleles can cause anthesis in wheat plants in short-day conditions, which is related to shortening the growing season and promoting wheat flowering; in contrast, the photoperiod-sensitivity allele requires long-day plants to grow for several weeks while being exposed to sunlight for more than 10 h a day before the onset of anthesis [16]. A study by Álvaro [17] on 12 Spanish and 12 Italian varieties showed that the varieties that carried the photoperiod-insensitivity alleles reached the anthesis stage eight and two days earlier, respectively, than those that carried the sensitivity allele.

Vernalization is controlled by four major genes: *Vrn-1*, *Vrn-2*, *Vrn-3*, and *Vrn-4* [18,19,20]. Among them, the dominant alleles, *Vrn-1*, *Vrn-3*, and *Vrn-4*, are related to the spring growth habit in wheat and promote its anthesis, while the dominant allele of *Vrn-2* is related to the winter growth habit in wheat and inhibits its anthesis [21]. Some studies have shown that *Vrn-1* has three homologous genes, *Vrn-A1*, *Vrn-B1*, and *Vrn-D1* [22]. Compared with *Vrn-B1* and *Vrn-D1*, *Vrn-A1* is the least sensitive to vernalization. The mutations within genes of these genes can cause an early anthesis date [23]. *Vrn-3*, comprising *Vrn-A3*, *Vrn-B3*, and *Vrn-D3,* is related to the expression of the FT protein and has a promoting effect on the anthesis date. A study on the vernalization genes of 47 wheat varieties in Australia showed that the effects of the genes *Vrn-A1* and *Vrn-B1* on the anthesis date were similar, but only varieties that carried the gene Vrn-B1 had an earlier anthesis date than those that carried a recessive gene [24].

The polymorphism of genes laid a foundation for studying the mechanism of response of wheat to anthesis-related alleles. The growing season–genotype interaction (GGI) has a significant effect on the anthesis date of different wheat varieties in specific areas. One of the key problems in the evaluation of different wheat varieties is assessing the influence of the GGIs on anthesis date, which would help select specific varieties for better adaptation [25,26]. Since the 1980s, tremendous efforts have been made by the International Maize and Wheat Improvement Center (CIMMYT) based on the breeding of different wheat genotypes on this interaction effect [27]. To adapt to more complex growing seasons, breeders select different varieties by studying their genotype structure, aiming to regulate the anthesis date of crops so that they are adapted to different climates and growing seasons of specific regions [28]. Previous studies have shown that the GGI effect has a greater impact on the growth of wheat when there are differences in growing seasons, especially in terms of temperature [29]. This interaction effect has been utilized by some researchers to improve the adaptability of varieties to various climatic conditions while constantly improving the adaptability of wheat varieties to local climatic conditions. For example, in Australia, farmers need to select the most appropriate wheat varieties for their local growing seasons to avoid high-temperature and drought stress during anthesis and ensure normal growth and anthesis [28,30].

The additive main effects and multiplicative interaction (AMMI) model is currently widely used in crop variety multipoint trials [31]. which is helpful for identifying genotypes adapted to agronomic traits and evaluating meteorological conditions in the study area [32]. AMMI can intuitively and effectively be used to analyze GGIs with the help of a biplot [33]. Rincent [34] used the AMMI model to analyze the factors affecting the phenological period and explained the proportion of growing seasons, genotypes, and GGIs. Studying the effect of GGIs can improve the adaptability of crops in target production areas, which is of great significance for wheat breeding. At present, numerous studies on the effect of growing seasons on anthesis date have been extensively studied. However, studies on the contributions of growing seasons, genotypes, and their interaction to wheat anthesis are still rare in the literature.

The objectives of this study were to (1) reveal changes in anthesis dates of varieties in different eras; (2) analyze the differences in the photoperiod and vernalization alleles in the process of variety replacement; (3) quantify the contribution of GGI on the anthesis date. The present study focused on the growing season temperature and photoperiod as integrated influential factors, as they are dominating environmental drivers affecting the anthesis date of wheat under sufficient light and irrigation.

## 2. Materials and Methods

### 2.1. Experimental Site and Growing Conditions

The study (2016–2019) was carried out at the Beijing Shunyi experimental station (40°15′ N, 116°55′ E) at the Institute of Environment and Sustainable Development in Agriculture of the Chinese Academy of Agricultural Sciences. This area has a temperate, semi-humid, continental monsoon climate. The sowing date is in the middle of October every year. The average accumulated temperatures from sowing to anthesis in 2016–2017 (16Y), 2017–2018 (17Y), and 2018–2019 (18Y) were 1261.95 d·°C, 1126.9 d·°C, and 1209.8 d·°C, respectively. The average annual temperature was 10–12 °C, average precipitation was approximately 640 mm, and the average day-length was approximately 2684 h over the three-year growing season. The soil at the experimental site was moist and sandy cinnamon soil. The irrigation treatment was consistent with the local fertile land.

### 2.2. Experimental Design and Plant Material

A randomized block design with three replications was used in the field experiment. Single row hand drill method was used for seed sowing with row-to-row and plant-to-plant spacing of 20 cm and 5 cm, respectively. The seeding depth was at 5 cm depth. The anthesis date was observed and recorded based on 50% of the middle spikelet flowering of wheat in the entire region [35]. The date of anthesis was transformed into sowing–anthesis total days (the first day after sowing was recorded as day 1) as the anthesis of wheat. Molecular genetics experiments were conducted to detect the dominant and recessive alleles of photoperiod and vernalization genes related to anthesis. The DNA of wheat seedling leaves were extracted by the high salt and low pH method [36], and photoperiod and vernalization gene loci were detected with the use of sequence-tagged sites (STSs) [14,20,37,38]. *Vrn-A1*, *Vrn-B1*, *Vrn-B3*, and *Ppd-B1b* were identified as recessive alleles for vernalization and photoperiod, respectively. Among the dominant alleles, *Vrn-D1* accounted for 85.9%, *Ppd-D1a* accounted for 69.7%, and *Ppd-A1a* accounted for 9.1%.

One hundred adaptive wheat varieties (including historical varieties from the 1940s and current varieties) came from the Chinese Crop Germplasm Resources Information System (https://www.cgris.net/; accessed on 25 August 2015). The list of the 100 wheat accessions used is not shown here due to the lack of permission. The eras and allele distribution of wheat varieties are shown in Figure 1. These varieties were divided into eight groups according to their eras of release (Figure 1). Among them, each of the groups 1950s, 1960s, and 1970s accounted for less than 10% of the total number of varieties. The 1940s, 1980s, 1990s, and 2010s varieties accounted for 10.0–20.0% of the total varieties, and the 2000s varieties accounted for 23.2% at most.

The photoperiod and vernalization alleles were combined in haplotypes in which A represented combinations of photoperiod alleles, and A1 (*Ppd-A1a* + *Ppd-D1a*). A2 (*Ppd-A1a + Ppd-D1b)*, A3 (*Ppd-A1b + Ppd-D1a*), and A4 (*Ppd-A1b + Ppd-D1b)* represented different photoperiod alleles, while B1 (*Vrn-D1*) and B2 (*vrn-D1*) represented different vernalization alleles. We included seven allelic combinations (Table 1). A3B2 and A4B2 accounted for the highest percentages (58.0% and 20.0%, respectively), and A1B1 accounted for the lowest percentage (1.0%).

As shown in Figure 2, the percentage of the combined genotypes A1B1, A1B2, A3B1, and A4B1 in the entire gene pool was low. A1B1 was distributed in the 1990s varieties, A1B2 in the 1970s and 1980s varieties, A3B1 in the 1990s–2010s varieties, and A4B1 in the 1940s–1950s varieties. A2B2 showed an increasing trend from the 1940s to the 1960s varieties but did not appear again in the 1970s and subsequent varieties. The presence of A3B2 increased from the 1940s to the 1980s varieties, while it decreased in the 1990s and 2010s varieties. A4B2 showed a decreasing trend from the 1940s to the 2000s varieties and did not appear in the 2010s varieties. The presence of *Ppd-D1a* increased between the 1940s and the 2010s varieties from 16% to 100%. The presence of *Ppd-D1b* started decreasing in the 1950s varieties, while it was completely absent in the 2010s varieties.

### 2.3. Data Analysis

The accumulated temperature data were based on the daily average temperature of the Shunyi base in Beijing to calculate the accumulated temperature from sowing to anthesis (an accumulated temperature ≥0 °C was referred to as accumulated temperature). To facilitate the annual accumulated temperature comparisons, the accumulated temperature was calculated from 17 October to 2 May of the following year. According to the daily meteorological data of the Beijing Meteorological Station from 1961 to 2010, the monthly average temperatures of these 50 years were calculated.

The anthesis dates from different growing seasons and genotypes were combined and subjected to analysis using the R language. The multiple comparison (“glht” function) was used to analyze the anthesis under different growing seasons or allelic combinations. Multi-way ANOVA (“aov” function) was used to analyze the effect of photoperiod alleles, vernalization alleles, and their interaction on anthesis. The Pearson correlation coefficient (“Performance Analysis” package) of anthesis date under different growing seasons was calculated, and the correlation coefficient was used as a rough estimation of heritability [39].

The AMMI model was built using the “agricolae” package in R language [29,40]. The AMMI model for the *g*th genotype in the *e*th environment is:

$$Y^{\mathrm{ger}}=\mu +\alpha^{g}+\beta^{e}+\overset{n}{\underset{n=1}{\sum }}\lambda^{n}\gamma^{\mathrm{gn}}\sigma^{\mathrm{en}}+\theta^{\mathrm{ger}}$$
where *Y^ger^* represents the *r*th repeated anthesis observation of the *g*th genotype in the eth growing season, *μ* is the overall average value, *α*^g^ is the deviation between the mean anthesis value and the total average value of each genotype, *β*^e^ is the deviation between the mean anthesis value and the total average value under each growing season, *λ*^n^ is the characteristic value of the interactive principal component analysis (IPCA), *γ*^gn^ and *σ*^en^ are the scores of the genotype and growing season of IPCA, *θ*^ger^ represents the error, and IPCA represents the principal component analysis value of the *r*th interaction of the *g*th genotype in the eth growing season.

The AMM1 biplot was drawn based on the interaction of the mean anthesis date and the first interactive principal component (IPCA1) to study early and late anthesis under different growing seasons. The AMMI2 biplot was drawn based on the IPCA1 and the second principal component interaction (IPCA2) to obtain the most adaptable genotype under different growing seasons, according to the distance between the genotype and the origin type combination.

## 3. Results

### 3.1. Response of Wheat Anthesis to Growing Seasons

The monthly mean temperature of the wheat growing season (from October to June of the next year) was plotted (Figure 3). In 16Y–18Y, the average temperature in 16Y was the highest, followed by 18Y, and that in 17Y was the lowest. Both the average temperature in the three growing seasons’ monthly average temperature of 16Y–18Y were higher than that in 1961–2010.

There was a significant correlation between anthesis date in different growing seasons and the increase in accumulated temperature, as anthesis occurred significantly earlier with the increasing temperature. The highest accumulated temperature in 16Y was 1262.0 °C. The accumulated temperature in 18Y was 1209.8 °C and that in 17Y was 1126.9 °C. The distribution of the anthesis date over the three growing seasons was uneven; it occurred the earliest in 16Y, followed by 18Y and 17Y, during which the anthesis onset was the latest (Figure 4A). The earliest onset of anthesis and the highest degree of variation was observed in 16Y, while 17Y had the lowest degree of variation. The Pearson correlation coefficient of the anthesis date among the different growing seasons was analyzed, and the heritability of 100 varieties was roughly estimated (Figure 4B). There was a significant correlation among the three growing seasons in terms of anthesis, indicating that the anthesis date was greatly affected by heritability and was relatively stable between years.

### 3.2. Response of Anthesis to Wheat Genotypes and Varieties Eras

The genotype had a significant effect on the anthesis date *(p* < 0.05). A3B2 resulted in significantly early anthesis, and A2B2 and A4B2 resulted in significantly late anthesis (*p* < 0.05). Throughout the study, the A1B1 genotype had different effects on anthesis (Figure 5B); in 16Y and 18Y, it resulted in early anthesis, while in 17Y, it resulted in late anthesis. A1B2 and A3B1 resulted in earlier anthesis than A4B1 and A2B2 *(p* < 0.05).

There were significant differences in the effects of photoperiod alleles, vernalization alleles, and their interaction on anthesis date in different growing seasons. As shown in Table 2, *Ppd-D1* had a significant impact on anthesis date throughout the entire study period (*p* < 0.05). In 16Y and 18Y, *Ppd-A1* had a significant effect on anthesis date (*p* < 0.05), while in 17Y, this effect was not significant *(p* > 0.05). *Vrn-D1* had a significant effect on anthesis date in 16Y *(p* < 0.05) but not in 17Y and 18Y *(p* > 0.05). The interaction effect of *Ppd-A1* and *Vrn-D1* was significant in 17Y and 18Y *(p* < 0.05), and the interaction effect of *Vrn-D1* and *Ppd-D1* was significant on the anthesis date throughout the entire study period *(p* < 0.05).

The anthesis date varied among different eras. As shown in Figure 5A, the anthesis date of the 1940s–1980s varieties gradually started occurring earlier; however, the differences in the anthesis date among varieties in the 1980s and later were not significant. Out of the total wheat varieties, 19.2% accounted for varieties from the 1940s and prior to that. In these varieties, the mean anthesis date was the highest and the degree of variation was large between 16Y and 17Y. The 1960s varieties accounted for the lowest percentage (3.0%), with a high degree of anthesis variation. The 1980s wheat varieties accounted for 14.1% of the total; in these varieties, the average anthesis date was the lowest between 17Y and 18Y. The 2010s varieties accounted for 23.2% of the total, and their degree of anthesis date variation was small. The average onset of anthesis of varieties from the 1940s to the 1980s started occurring progressively earlier, and the variation in the onset of anthesis from the 1980s to the 2010s was small.

### 3.3. Response of Wheat Anthesis to the GGI

The GGI had a significant effect on anthesis date (*p* < 0.05). The photoperiod and vernalization alleles formed combined genotypes (Table 2). The AMMI model was constructed according to the growing seasons and genotypes (Table 3, Figure 6). The results showed that the growing seasons accounted for the greatest effect on anthesis date (66.3%), followed by the genotypes (21.9%) and the interaction of the two variables (6.5%). The two main interaction effect components, IPCA1 and IPCA2, accounted for 100.0% of the total effect on the anthesis variation; IPCA1 and IPCA2 accounted for 94.6% and 5.4% of the variation, respectively. The F value of the two components reached a significant level (*p* < 0.05). The variance due to the residual was only 1.4%.

There were differences in the proportions of the anthesis date of wheat varieties in different eras affected by GGIs. As the genotypes A2B2 and A4B2 were mainly distributed in the 1940s, 1950s, and 1960s, with a delaying effect on the anthesis date, only two genotypes of the 1960s varieties were found, which did not meet the requirements of the AMMI model variance analysis. Therefore, this study only conducted an analysis of variance on the anthesis date of varieties in the 1940s and 1950s (Table 4). Among the varieties in the 1940s, the effects of the growth season, genotype, and their interaction on the anthesis date were very significant (*p* < 0.001). Among them, genotype had the greatest effect on anthesis date (70.6%), followed by growth season (25.0%), and the proportion of interaction effect was the smallest (3.2%). In the 1950s, growth season, genotype, and their interaction had significant effects on the anthesis date (*p* < 0.05). Among them, the growing season had the greatest effect on anthesis (93.4%), followed by genotype (4.2%), and the proportion of the interaction effect was the smallest (1.6%).

## 4. Discussion

The growing seasons significantly affected the distribution of wheat phenology, interfering with not only the timing of each phenophase but also with the growing seasonal distribution of the same phenophase [41]. In this study, Shunyi was selected as the study site for a three-year experiment. At present, the research on anthesis in China is mostly concentrated in the Huang Huai Hai region, while the research in North China is still rare in the literature. The results showed that the anthesis date occurred early in the year with the high accumulated temperature, which is consistent with the results of other studies [6,42]. In this study, the variation in the anthesis dates of 100 varieties increased with the increasing temperature, which indicated that the response of anthesis date to temperature differed among varieties. In this regard, such information can help to select wheat varieties that favor anthesis under target temperature scenarios.

To adapt change to abiotic and biotic stresses and increase wheat production, wheat varieties are constantly being replaced, and the distribution of photoperiod and vernalization alleles differs among different wheat varieties [2]. The varieties selected in this study spanned eight eras and included some landraces. The landraces contained rich wheat vernalization gene types (a few varieties had some deletions in the Vrn and Ppd genes), which increases genetic diversity and is beneficial for studying the response of anthesis date to genotypes [43]. Our results showed that the anthesis of varieties dating before the 1980s was gradually early, while the anthesis of varieties dating after this era was relatively stable. Apart from global warming, this phenomenon could also be explained by the fact that Chinese producers started selecting varieties with high temperature requirements in the 1980s, thereby slowing down the phenological shortening caused by increases in the temperature and contributing to the inhibition trend of early anthesis [44]. The study by Yang [45] on *Ppd-D1a* showed that by the 1970s, this allele was present in most varieties in North China. The results of the present study corroborate those of Yang [45]. Our results showed that varieties of different eras have been evolving since the 1940s; additionally, the proportion of *Ppd-D1a* increased from 16% in the 1940s to 100% in the 2010s. The proportion of A4B2 decreased between the 1940s and the 2000s, but this genotype did not appear in the 2010s. The proportion of A3B2 increased from the 1940s to the 1980s, while it decreased in the 2010s. The results obtained here are similar to those presented previously [46]. To improve the yield in the 1980s, the varieties with weak cold hardness (early maturing varieties) were selected with early anthesis. However, owing to the continuous frost damage, the varieties with weak cold hardness were used less in the 2010s. This study only examined the response of varieties in different eras to accumulated temperature. However, a more detailed study of the growing seasons affecting wheat anthesis is needed, as this will allow adjustment for precipitation, light, and other factors affecting wheat anthesis.

The diversity of photoperiod and vernalization allelic combinations is the key component to controlling phenology, which regulates the anthesis date [47,48,49]. The effect of the photoperiod allele and vernalization allele on the anthesis date is summarized from literature in Table 5. Eagles [50] combined the photoperiod and vernalization alleles to study the effect of gene combinations on heading date. Their results showed that compared with a single gene, the combined genotype more comprehensively explains the variation of heading date among different varieties. In the present study, the photoperiod and vernalization alleles were combined to study the effect of genotypes on the anthesis date. A4B2 resulted in delayed anthesis because it carries *Ppd-A1b*, *Ppd-D1b*, and *Vrn-D1*, all of which also had a delaying effect on the anthesis date. The A1B1 genotype consisted of alleles, such as *Ppd-A1a*, *Ppd-D1a*, and *Vrn-D1*, which resulted in early anthesis; however, A1B1 resulted in delayed anthesis in 17Y, which was due to the small number of varieties in which the A1B1 genotype was expressed and the large fluctuation influenced by the growing seasons. The A3B2 genotype consisted only of *Ppd-D1a*, which can promote anthesis. However, as *Ppd-D1* had a significant effect on the anthesis date and its photoperiod insensitivity was greater than that of *Ppd-A1* [16], it resulted in a significantly earlier anthesis. The A1B2 and A3B1 genotypes resulted in earlier anthesis than the A4B1 and A2B2 genotypes; this corroborates the results of a study on heading stage [51], because A1B2 and A3B1 possessed two alleles (including *Ppd-D1a*), while A4B1 and A2B2 possessed only one allele related to early anthesis. Our results showed that the A1B1 genotype had earlier anthesis than A1B2, while the A3B1 genotype resulted in earlier anthesis than the A3B2 genotype. This indicated that *Vrn-D1* affected the anthesis date, as the interaction of *Vrn-D1* with *Ppd-A1* and *Ppd-D1* had a significant effect on the anthesis date. This study only examined the photoperiod and vernalization allelic in terms of genes; however, other non-allelic genes, *VRN1* and *PPD1*, that affect the florescence of wheat have not been studied in this paper, which needs to be further studied in the future.

The anthesis date was affected by the interaction of complex genes and growing seasons. The regulation of the wheat photoperiod alleles on the anthesis date was related to the length of the day, and the effect of the vernalization loci on the anthesis date was affected by the overwintering temperature [52,53]. In this study, *Ppd-A1* had a significant effect on the anthesis date in 16Y and 18Y but not in 17Y. This could be due to the low photoperiod sensitivity of *Ppd-A1* and the climate characteristics associated with low temperatures and long days in China [54]. Under growing seasons with lower accumulated temperature and longer day-length in in 17Y, both photoperiod-sensitive and -insensitive alleles could contribute to wheat flower formation normally; therefore, the difference in anthesis date was not significant. The composition of the vernalization alleles was related to the meteorological condition of wheat growth. *Vrn-A1*, *Vrn-B1*, and *Vrn-B3* were all expressed in wheat possessing a winteriness growth habit with only *Vrn-D1* had a different performance. *Vrn-D1* had a significant effect on the anthesis date in 16Y (*p* < 0.05) but not in 17Y and 18Y (*p* > 0.05), which was probably because vernalization is limited by temperature or chilling requirement [11]. In 17Y and 18Y, wheat varieties could cope with winter temperatures with no significant difference in anthesis date.

Qualitative analysis of GGIs on the anthesis date of crops could facilitate the selection for adaptative germplasm in breeding programs. For example, Li evaluated the influencing factors of sorghum anthesis dates throughout seven growing seasons (three locations and different years) in the United States based on GGIs [39]. Among them, the effects of growing seasons, genotypes, and interaction accounted for 63.0%, 5.0%, and 20.5%, respectively. The results were helpful for understanding the variation in anthesis, revealing the regulation mechanism of the anthesis date, and they provided the possibility to find suitable flowering crop varieties in the study area. In our study, the AMMI statistical model was selected to study the anthesis date (varieties in 1940s and 1950s). Growing seasons had the greatest influence on anthesis date (66.3%), followed by the genotype (21.9%) and the interaction of the two variables (10.1%). Therefore, the growing seasons of specific areas should be prioritized when choosing an area to breed wheat; the second factor that should be considered is the genotype of the wheat. The interaction effect obtained from our study was relatively small compared to that obtained by Li [39], which is probably because the fluctuation among growing seasons in our study was smaller than that in different places [32]. Another plausible reason is probably because of the differences in the C4 (sorghum) and C3 (wheat) plants [55]. The IPCA1 and IPCA2 of the AMMI model explained 94.6% and 5.4% of the interaction effects on the anthesis date, respectively. This result indicates that meteorological and non-growing seasonal factors accounted for 94.6% and 5.4% of the interaction effects, respectively. Two principal components, IPCA1 and IPCA2, explained 100% of the interaction effect on anthesis date variation, which was similar to the results obtained by Gabriel [56], who explained 100% (IPCA1 + IPCA2) of the interaction effect on the cotton yield variation.

Understanding the performances’ of varieties is helpful for providing a basis for evaluating the adaptability of specific varieties. In this study, the contribution of growing seasons, genotypes, and their interaction to the anthesis date was essential for a scientific work. Based on our results, we can conclude that growing seasons contributed tremendously to the anthesis date of wheat. In future wheat breeding, more consideration should be given to growing seasons, and the gene combination with the strongest adaptability to the growing seasons should be selected. The variance analysis was carried out on the anthesis date of wheat varieties in different eras. Compared with genotypes released in other eras, it was found that genotype was the main factor affecting the anthesis date of the 1940s varieties (landraces), accounting for 70.6% of the anthesis date and 25.0% of the growing seasons, which indicates that this genotype was more stable than the new genotypes and more affected by heritability.

The genotypes that have a suitable anthesis date in a specific region should be selected, as these help alleviate the adverse effects of growing seasons on wheat growth. Our study took place in the North China Plain. Some studies have shown that the climate of this region will continue to become warmer in the future [57,58], followed by the shortening of the wheat growth cycle [59]. Liu studied wheat phenology in the North China Plain under global warming conditions [60]; their results showed that warming mainly occurs in the vegetative growth stage of wheat (before anthesis), leading to the shortening of the vegetative stage, which, because this is an important phase/time for N uptake, is later re-mobilized during the linear grain filling phase; marked differences among genotypes are known. Considering this situation, Li pointed out that the length of the wheat vegetative growth stage should be maintained to ensure the number of grains per spike [61]. To avoid the influence of climate warming and shortening phenological period on yield, late anthesis varieties were selected [62]. In the present 2study, GGIs had a significant effect on the anthesis date of all the 100 wheat varieties, whereas the influence of growing seasons dominated. However, when the 1940s cultivars were analyzed separately, it was found that the genotypes had a great influence on the anthesis date. Therefore, the varieties of the 1940s tended to have more potential to be used for developing resilient genotypes. This is in line with directions for breeding under climate warming conditions that select late flowering and mature varieties to cope with climate warming [63]. The present results showed that the A2B2 and A4B2 genotypes delayed anthesis date under the growing seasonal conditions among the three years, and the A4B2 genotype was mainly distributed in the 1940s varieties, which were greatly influenced by the genotypes. Therefore, from an application perspective, the 1940s varieties can be useful for breeding varieties to avoid premature anthesis in future breeding plans.

## 5. Conclusions

In summary, this study showed that the response of the anthesis date to the photoperiod and vernalization alleles varied among eras in the process of variety replacement. The effects of growing seasons, genotypes, and their interactions on the anthesis date of winter wheat was actually described by the AMMI model and double plot analysis. We suggest that, in future breeding plans, the 1940s varieties can be used as parents for breeding new varieties with a delayed effect on anthesis, which could facilitate the breeding of resilient adaptive wheat varieties to cope with climate warming in the future.

## Figures and Tables

**Figure 1 biology-10-00955-f001:**
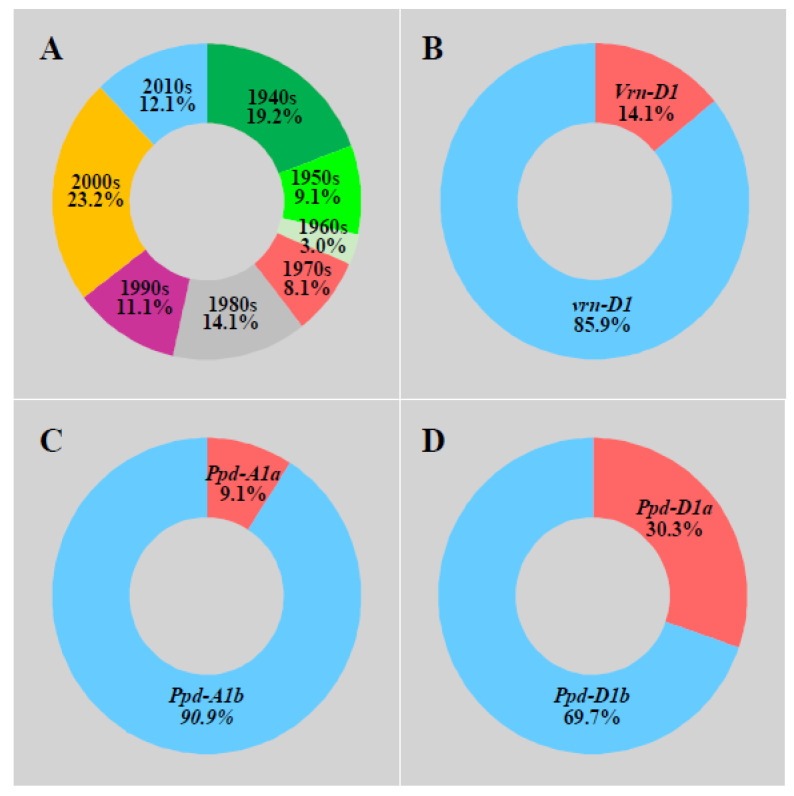
Eras and allele distribution of wheat varieties. Note that the 1940s represent the varieties dating during and prior to the 1940s. (**A**) 100 wheat varieties were divided into eight groups according to their eras of release. (**B**) 100 wheat varieties were divided into the *vrn-D1* allele. (**C**) 100 wheat varieties were divided into the *Ppd-A1* allele. (**D**) 100 wheat varieties were divided into the *Ppd-D1* allele.

**Figure 2 biology-10-00955-f002:**
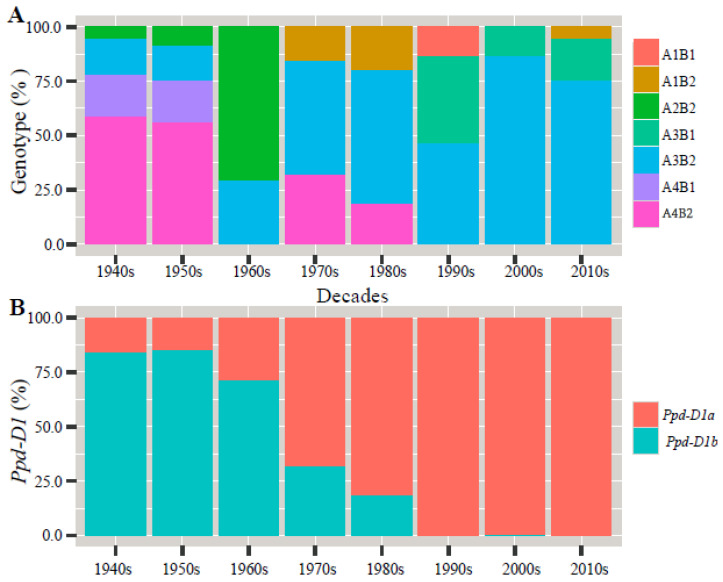
Percentage column plots of genotype and photoperiod genes in different eras for 100 wheat varieties evaluated in three growing seasons: (**A**) the percentage of the combined genotypes to their eras of release; (**B**) the percentage of the *Ppd-D1* allele to their eras of release.

**Figure 3 biology-10-00955-f003:**
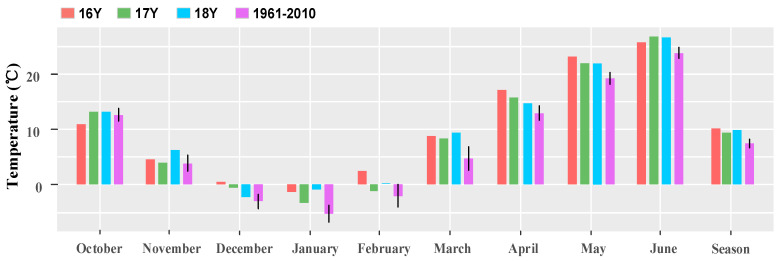
Bar chart of the monthly average temperature of wheat in 16Y–18Y and the last 50 years (1961–2010). 16Y is from October 2016 to June 2017; 17Y is from October 2017 to June 2018; 18Y is from October 2018 to June 2019; 1961–2010 is the monthly average temperature from October 1961 to June 2010. Growing season is the average temperature from October to June of wheat in the whole phenophase.

**Figure 4 biology-10-00955-f004:**
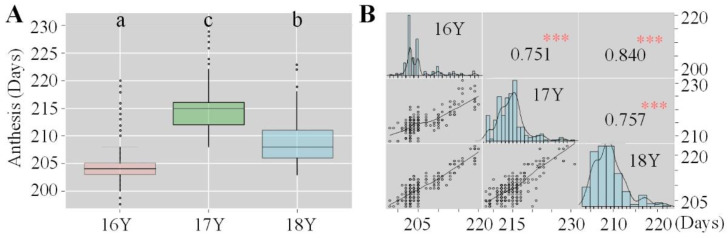
Boxplot and correlation matrix of the anthesis days of 100 wheat varieties evaluated in three growing seasons: (**A**) boxplot of the anthesis of 100 wheat varieties evaluated in three growing seasons; (**B**) correlation matrix of the anthesis of 100 wheat varieties evaluated in three growing seasons. 16Y, 2016–2017 growing season; 17Y, 2017–2018 growing season; 18Y, 2018–2019 growing season. The different letters (a, b, and c) represent significance at the 0.05 level. *** Significant at the 0.001 level; ** significant at the 0.01 level; * significant at the 0.05 level.

**Figure 5 biology-10-00955-f005:**
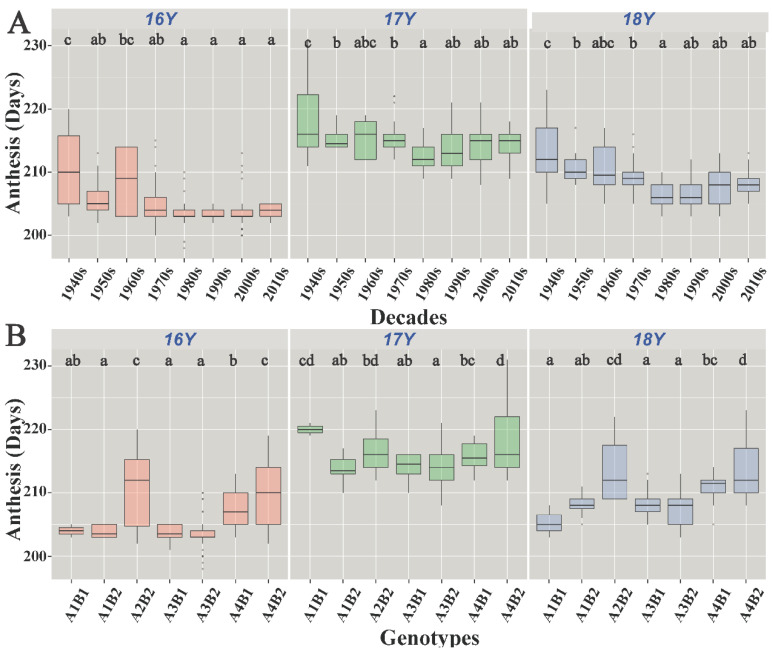
Boxplot of anthesis: (**A**) boxplot of anthesis of varieties of different eras; (**B**) boxplot of different genotypes affecting anthesis. 16Y, the 2016–2017 growing season; 17Y, the 2017–2018 growing season; 18Y, the 2018–2019 growing season. The different letters indicate significance at the 0.05 level.

**Figure 6 biology-10-00955-f006:**
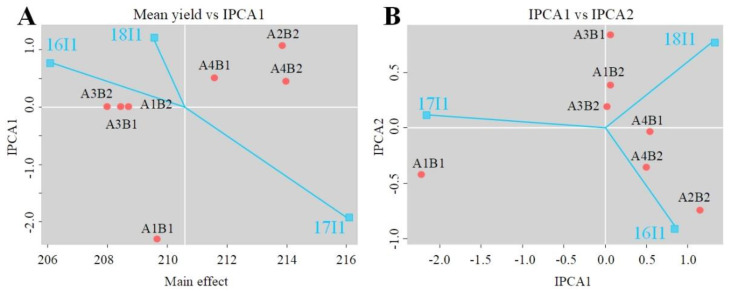
AMMI1 (**A**) and AMMI2 (**B**) biplot for wheat anthesis of 100 wheat varieties evaluated in three growing seasons.

**Table 1 biology-10-00955-t001:** Photoperiod and vernalization allelic combinations of 100 winter wheat varieties evaluated in three growing seasons (2016–2019).

Genotype	Photoperiod and Vernalization Genotype	Percentage (%) ^†^
A1B1	*Ppd-A1a + Ppd-D1a + Vrn-D1*	1.0
A1B2	*Ppd-A1a + Ppd-D1a + vrn-D1*	4.0
A2B2	*Ppd-A1a + Ppd-D1b + vrn-D1*	4.0
A3B1	*Ppd-A1b + Ppd-D1a + Vrn-D1*	7.0
A3B2	*Ppd-A1b + Ppd-D1a + vrn-D1*	58.0
A4B1	*Ppd-A1b + Ppd-D1b + Vrn-D1*	6.0
A4B2	*Ppd-A1b + Ppd-D1b + vrn-D1*	20.0

^†^ The percentage distribution of genotypes in 100 wheat varieties.

**Table 2 biology-10-00955-t002:** Variance analysis of the effects of photoperiod alleles, vernalization alleles, and their interactions on the anthesis of 100 wheat varieties evaluated in three growing seasons (2016–2019).

Growing Seasons	Source of Variance	Sum Sq	Mean Sq	F Value	Pr > F
16Y	*Ppd-A1*	76.8	76.8	8.3	0.004
	*Ppd-D1*	2368.7	2368.7	256.7	<0.001
	*Vrn-D1*	40.7	40.7	4.4	0.037
	*Ppd-A1:Ppd-D1*	3.4	3.4	0.4	0.546
	*Ppd-A1:Vrn-D1*	2.1	2.1	0.2	0.635
	*Ppd-D1:Vrn-D1*	52.1	52.1	5.7	0.018
17Y	*Ppd-A1*	13.7	13.7	1.3	0.249
	*Ppd-D1*	1003.7	1003.7	98	<0.001
	*Vrn-D1*	9.5	9.5	0.9	0.336
	*Ppd-A1:Ppd-D1*	20.6	20.6	2	0.158
	*Ppd-A1:Vrn-D1*	42.8	42.8	4.2	0.042
	*Ppd-D1:Vrn-D1*	73.5	73.5	7.2	0.008
18Y	*Ppd-A1*	42.2	42.2	5.2	0.024
	*Ppd-D1*	1877.2	1877.2	230.2	<0.001
	*Vrn-D1*	9.8	9.8	1.2	0.274
	*Ppd-A1:Ppd-D1*	2.3	2.3	0.3	0.596
	*Ppd-A1:Vrn-D1*	12.2	12.2	1.5	0.222
	*Ppd-D1:Vrn-D1*	91.9	91.9	11.3	0.001

16Y, the 2016–2017 season; 17Y, the 2017–2018 season; 18Y, the 2018–2019 season. Degrees of freedom (df) = 1 for all presented calculations.

**Table 3 biology-10-00955-t003:** AMMI analysis of variance on the anthesis of 100 wheat varieties evaluated in three growing seasons (2016–2019).

Source of Variance	df	Sum of Squares	Fischer’s Ratio	% of Variance
Growing seasons (S)	2	1047.7	347.8 ***	66.3
Genotype (G)	6	345.5	99.1 ***	21.9
S:G	12	158.9	11.6 ***	10.1
IPCA1	7	150.3	35.2 ***	94.6
IPCA2	5	8.6	2.8 *	5.4
Residuals	36	22.0		1.4

*** Indicates significance at the 0.001 level, ** Indicates significance at the 0.01 level, and * indicates significance at the 0.05 level. IPCA1, first interactive principal component; IPCA2, second interactive principal component.

**Table 4 biology-10-00955-t004:** Variance analysis on the effect of growing season–genotype interactions (GGIs) on the anthesis of wheat varieties in different eras in 16Y–18Y.

Variety Eras	Source of Variance	Df	Sum of Squares	Fischer’s Ratio	% of Variance
1940s	Growing seasons (S)	2	288.8	212.5 ***	25.0
	Genotype (G)	3	816.4	333.3 ***	70.6
	S:G	6	37.3	7.6 ***	3.2
	Residuals	18	14.7		1.2
1950s	Growing seasons (S)	2	645.1	265.3 ***	93.4
	Genotype (G)	3	28.9	31.6 ***	4.2
	S:G	6	11.1	6.1 **	1.6
	Residuals	18	5.5		0.8

*** Indicates significance at the 0.001 level, ** indicates significance at the 0.01 level, and * indicates significance at the 0.05 level. The 1960s variety was not analyzed because it did not meet the analysis of variance of the AMMI model.

**Table 5 biology-10-00955-t005:** Regulatory effects of photoperiod and vernalization alleles on anthesis.

	Gene	Allele	Effect	Reference
Vernalization gene	*Vrn-D1*	*Vrn-D1* (Spring growth habit)	Promoted flowering	[23]
		*vrn-D1* (Winter growth habit)	Delayed flowering	
Photoperiod genes	*Ppd-A1*	*Ppd-A1a* (Photoperiod-insensitive gene)	Promoted flowering	[17]
		*Ppd-A1b* (Photoperiod-sensitive gene)	Delayed flowering	
	*Ppd-D1*	*Ppd-D1a* (Photoperiod-insensitive gene)	Promoted flowering	
		*Ppd-D1b* (Photoperiod-sensitive gene)	Delayed flowering	

## Data Availability

The data presented in this study are available on request from the corresponding author.

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
