# Peer review of "Understanding the Effects of Growing Seasons, Genotypes, and Their Interactions on the Anthesis Date of Wheat Sown in North China"

_biology, 2021, doi:10.3390/biology10100955_

Round 1
Reviewer 1 Report
The authors used an AMMI model to investigate the contribution of several photoperiod and vernalization alleles as well as their interaction with the environment (e.g. temperature) on the wheat anthesis date. Overall, the manuscript is well-written and timely, given the predictions of the increasing global temperatures. However, there is a major caveat that needs to be addressed before the manuscript is suitable for publication.
- There are several mistakes in English in the manuscript. E.g., the sentence "At present, numerous studies have conducted on the influence of growing seasons on anthesis date has been extensively studied" in L110-111 doesn't make sense, though I can understand what the authors wanted to say from the context. However, these mistakes are rare enough so that the manuscript is clearly readable, and the overall English is quite adequate. I would suggest that the authors carefully go over their manuscript again as these mistakes appear to be leftovers from the revising the manuscript for clarity and typos rather than the lack of English proficiency and, therefore, are easily fixable by the authors themselves.
- My primary concern with the results are the inconsistency of the effect of the alleles depending on the growing season. E.g., "In 16Y and 18Y, Ppd-A1 had a significant effect on anthesis date (P <0.05), while in 17Y, this effect was not significant (P > 0.05).", Lines 252-253. While the authors came up with a reasonable explanation in the discussion section, this made me wonder whether the AMMI model was applied properly (and if it was applicable at all) to the data. While I'm not familiar with the model firsthand, my understanding is that in certain cases the error term θger should follow normal distribution if the model's assumptions were correct. Have the authors checked it or have done any other quality control to verify that this dramatic effect of the growing season has indeed biological origin, rather than being a modeling artifact?
- The major caveat I mentioned above, is that the explanation for the reason of the inconsistency for the effects of the Ppd-A1 allele (Lines 386-392) depending on the growing season, directly contradicts both Table 2 and the aforementioned lines 252-253. It states that Ppd-A1 had significant effect in 16Y and not 17Y, then uses the meteorological data of 16Y to present an explanation. Unless, I have somehow misunderstood these passages, then either a) The authors have a typo in Table 2 and the lines 252-253, b) the model works for these dates but the reason for this allele inconsistency is unknown, c) The model cannot fit well or be applied to the Ppd-A1 allele, or d) Ppd-A1 allele does not significantly affect the anthesis date and these results are a modeling artifact. This concern needs to be addressed.
Author Response
- There are several mistakes in English in the manuscript. E.g., the sentence "At present, numerous studies have conducted on the influence of growing seasons on anthesis date has been extensively studied" in L110-111 doesn't make sense, though I can understand what the authors wanted to say from the context. However, these mistakes are rare enough so that the manuscript is clearly readable, and the overall English is quite adequate. I would suggest that the authors carefully go over their manuscript again as these mistakes appear to be leftovers from the revising the manuscript for clarity and typos rather than the lack of English proficiency and, therefore, are easily fixable by the authors themselves.
Response: We accept this concern and have made the corresponding revisions. For example, L87, L124, L140, L164, L205 and L420 in revised version.
- My primary concern with the results are the inconsistency of the effect of the alleles depending on the growing season. E.g., "In 16Y and 18Y, Ppd-A1 had a significant effect on anthesis date (P <0.05), while in 17Y, this effect was not significant (P > 0.05).", Lines 252-253. While the authors came up with a reasonable explanation in the discussion section, this made me wonder whether the AMMI model was applied properly (and if it was applicable at all) to the data. While I'm not familiar with the model firsthand, my understanding is that in certain cases the error term θger should follow normal distribution if the model's assumptions were correct. Have the authors checked it or have done any other quality control to verify that this dramatic effect of the growing season has indeed biological origin, rather than being a modeling artifact?
Response: We agree with concern. We used Multi-way ANOVA (“aov” function) to analyze the effect of photoperiod alleles, vernalization alleles, and their interaction on anthesis. We used the AMMI model to understand effects of growing seasons, genotypes, and their interactions on the opposition date.The standard AMMI model is not appropriate in all cases. The AMMI model needs to observe all GE combinations (i.e. the two-way data sheet must be complete and all observations in the two-way data sheet have the same weight in the final model) (Paderewski et al., 2018). Our data meet this condition. And in present study, two principal components, IPCA1 and IPCA2, explained 100% of the interaction effect on anthesis date variation. Based on these, this paper correctly uses the data for AMMI model analysis. For this concern, we updated relevant content regard to robustness of model use.
- The major caveat I mentioned above, is that the explanation for the reason of the inconsistency for the effects of the Ppd-A1 allele (Lines 386-392) depending on the growing season, directly contradicts both Table 2 and the aforementioned lines 252-253. It states that Ppd-A1 had significant effect in 16Y and not 17Y, then uses the meteorological data of 16Y to present an explanation. Unless, I have somehow misunderstood these passages, then either a) The authors have a typo in Table 2 and the lines 252-253, b) the model works for these dates but the reason for this allele inconsistency is unknown, c) The model cannot fit well or be applied to the Ppd-A1 allele, or d) Ppd-A1 allele does not significantly affect the anthesis date and these results are a modeling artifact. This concern needs to be addressed.
Response: We accept this concern and have made the corresponding revisions. We have a typo in the lines 386-392, which has been revised accordingly. The statistical results used in L386-392 are extremely significant results (P < 0.001), which have been modified into a unified standard with table 2and L255-253 (P < 0.05). Thanks for correcting us.
Reviewer 2 Report
Quantitative studies on the effects of growing season, genotype and their interaction to anthesis date of wheat are helpful to provide scientific reference for selecting or developing adaptive varieties in target environments. For this, additive main effect and multiplicative interaction (AMMI) model and GGI biplot analysis were used. A large portion of the total variation (of 100 genotypes from different eras) was explained by growing seasons (66.3%), while genotypes and GGI explained 21.9% and 10.1% of wheat-anthesis date. Varieties from the 1940’s and before had a great influence on the anthesis date, suggesting these germplasms tend to avoid premature and could facilitate developing phenological resilient varieties.

Author Response
Quantitative studies on the effects of growing season, genotype and their interaction to anthesis date of wheat are helpful to provide scientific reference for selecting or developing adaptive varieties in target environments. For this, additive main effect and multiplicative interaction (AMMI) model and GGI biplot analysis were used. A large portion of the total variation (of 100 genotypes from different eras) was explained by growing seasons (66.3%), while genotypes and GGI explained 21.9% and 10.1% of wheat-anthesis date. Varieties from the 1940’s and before had a great influence on the anthesis date, suggesting these germplasms tend to avoid premature and could facilitate developing phenological resilient varieties.
(1) Prunus mume! There are many other and actual references from wheat in the literature (e.g. Slafer etc.)
Response: We accept this concern and have made the corresponding revisions.
(2) L42: Response: Agreed and it has been modified to "phenological"
(3)L43: mechanism not the right terminology
Response: We accept this concern and have made the corresponding revisions.
Could also see our response to another reviewer.
(4)L75: Response: Agreed and revised.
(5)L138: Response: Agreed and revised.
(6) L149: Response: Agreed and revised.
(7) L178: The, normal use are 30 years
Response: Because the varieties selected in this study have a large age span, meteorological conditions also use more year data to make the temperature changes in different months more in line with the research needs of this paper.
(8)L188: “Percentage is the percentage distribution of genotypes in 100 wheat varieties.’ Improve
Response: Agreed and revised.
(9)L266: Response: Agreed and revised. It has been modified to 1126.9℃.
(10)L236-L238: Response: Agreed and revised.
(11)L240: Response: Agreed and revised. It has been modified to anthesis days.
(12)L273: Response: Agreed and revised.
(13)L291: Response: Agreed and revised.
(14)L349: Response: Agreed and revised.
(15)L352: Response: Agreed and revised.
(16) L358: Response: Agreed and revised.
(17)L390: Response: Agreed and revised.
(19)L397: Response: Agreed and revised.
(20)L407: Response: We agree with this concern. We add content in this paragraph.
(21)L409: Response: Agreed and revised.
(22)L426 Response: Agreed and revised.
(23) L443: Response: Agreed and revised.
Reviewer 3 Report
The manuscript ‘Biology-1385222’ deals with very interesting and important topic of Time to anthesis (or flowering) trait in relation to genotypes and growing seasons in wheat. The manuscript is written in logical style, with good Introduction, appropriate M&M. important Results and interesting Discussion. Therefore, the manuscript can be conditionally accepted with a subject to minor revision. Authors have to address major comment and minor corrections indicated below carefully and properly to make their manuscript ready for the acceptance and publishing.
Major comment:
(1) Authors must clarify that they used only winter but not spring wheat in their study. This is particular clear in L125 and L217 that plants were grown from October over winter and harvested in June. Spring wheat cannot grow in such vegetation period. However, not only winter wheat but also ‘facultative’ wheat can be grown overwinter. Facultative wheat accessions are those that can be used either as winter or as spring wheat. Therefore, in fact, 100 studied wheat cultivars might represent only winter wheats or a mixture of both winter and facultative wheats. Authors must clarify the developmental type or types of used wheat germplasms. If all 100 studied wheat accessions were winter only, it will be relatively simple and minor corrections, ‘winter wheat’ must be inserted, at least, in the Title (‘…anthesis date of winter wheat sown…’), in Abstract, Keywords, at the first occurrences in Introduction and in the Discussion, and it has to be carefully described in M&M section.
However, the situation can be transformed into major changes, if at least one (or more) from 100 studied wheat accessions were facultative, and such wheat accessions had either winter or spring type of development. In this case, Authors must split and show their analyses separately, for winter and facultative wheats. This is not ‘a tragedy’ but it is very important to provide and publish correct and absolutely clear information, avoiding any potential misleading of Readers. If both winter and facultative wheats were used, the Title has to be modified, for example as follows: ‘Understanding effects of growing seasons, genotypes, and their interactions on the anthesis date of winter-growing wheat in North China’. In this case, ‘winter-growing’ means a mixture of both winter and facultative accessions of wheat. Following changes must be included in the text accordingly but additional analyses of two groups (winter and facultative wheats) and their comparison will be required.
Minor corrections/notes:
(2) L10-11, L69-77. Authors must be careful and indicate Latin names of plants species and names of genes in Italics: ‘Triticum aestivum’; ‘Vrn-1, Vrn-2, Vrn-3, and Vrn-4’ genes; and so on.
(3) L20, L24, L50, L244, L261, L321, L356, L384, L387, L460 and maybe in other parts of the manuscript. The term “wheat-anthesis date” is unclear, particular with using of a hyphen between words ‘wheat’ and ‘anthesis’. What does this mean and what Authors want to say? Starting from the Title and Abstract and through entire manuscript, Author also use correct term ‘anthesis date’, which is a synonym of ‘flowering date’, and it means ‘number of days until anthesis or flowering’. Authors have to correct all terms ‘wheat-anthesis date’ in the manuscript.
(4) L34-35 and L328. The term ‘biological stress’ or ‘biological stressor’ is non-typical. It is better to change it for ‘biotic stress’. In all cases Authors have to use more popular terms ‘abiotic and biotic stresses’.
(5) L42. Mistyping ‘phenological’.
(6) L43-44. Please re-phrase the sentence: “Crop breeding is a key mechanism for adaptation of cropping systems to local climate”. Crops breeding cannot be a mechanism… ‘Crop breeding’ is the ‘science’ or ‘scientific direction’, which studies (or based on) adaptation of cropping systems…
(7) L57-58. Authors used incorrect statement: “…the homoelogous gene of three types of Ppd-1, namely Ppd-A1, Ppd-B1, and Ppd-D1”. It has to be changed for clear meaning. The proposed variants are as follows ‘…three homoelogous genes, Ppd-A1, Ppd-B1, and Ppd-D1’ or ‘…Ppd-1 gene with copies in three homeologous chromosomes in A, B and D genomes’.
(8) L62. The term “short-day plants” is unclear, particular for wheat. It is better to change it for better meaning as follows: ‘wheat plants in short-day conditions’ or ‘plants grown in short-day environment’.
(9) L78-81. The sentence starting from: “A study on the vernalization genes…” with the reference to [Harris et al., 2017], is unclear. Please split it for two sentences and explain better, what is happened in wheat with dominant and recessive alleles of both Vrn-A1 and Vrn-B1 genes and with isolated Vrn-B1 for time to anthesis.
(10) L124-125. The following phrase is unclear: “…a semi-moist temperate continental monsoon climate”. What does ‘semi-moist temperate’ means? Is it cold or hot? Please clarify.
(11) L134. What was a density of wheat plants? The phrase “from 200 cm length and 25 cm spacing” does not help. Please provide information for both: ‘number of plants in one metre of row (or distance between plants in a row)’ and ‘average number of plants per m2 (or another density characteristic)’.
(12) L139. The phrase “recessive expression of photoperiod” represents a poor laboratory jargon. Both terms ‘recessive’ and ‘expression’ have very strong definitions in genetics, where ‘recessive’ and ‘dominant’ alleles only are used. Similarly, ‘expression’ is used only for ‘gene expression’ as activation of mRNA production during onthogenesis or in response of plants to external action. Please modify for clear meaning, what Authors want to say with this phrase.
(13) L143-145. Please cut a part of the second sentence making it shorter and more clear, combining two parts together. Original: “The vernalization and photoperiod alleles were identified. The alleles of Vrn-A1, Vrn-B1, Vrn-B3, and Ppd-B1 (Vrn-A1, Vrn-B1, Vrn-B3, and Ppd-B1b, respectively) were recessive”. Proposed: ‘Vrn-A1, Vrn-B1, Vrn-B3, and Ppd-B1b were identified as recessive alleles for vernalization and photoperiod, respectively’.
(14) L147. Authors must provide a full list of 100 used wheat accessions, for example in the Supplementary Materials. Moreover, this list has to include elementary information for each wheat cultivar: Name, Registration number, Origin (Originator), Year of release, Pedigree, One or two sentences of brief description of main characteristics for each wheat cultivar. If this is not acceptable for Authors, please insert here your statement as follows: ‘The list of 100 used wheat accessions represents the confidential information and cannot be released in open publicly-available papers due to … [insert your explaination, why the list cannot be provided]’. Only with the list of 100 used wheat accessions, it is possible (at least theoretically) to reproduce the presented results.
(15) L148-149. Please provide a reference for ‘Chinese Germplasm Resource Bank’. Web-site link would be very good but Authors have to be sure that web-site has English version for Readers.
(16) L151-164 and Figure 2A, and L179-185 and Table 1. Authors have to think and re-arrange the order of these two fragments with Figure 2A and Table 1. This is because results for seven genotypes (A1B1-A4B2) were shown and described earlier in sub-section 2.2 but their definition and meaning were provided later, in sub-section 2.3 (which was incorrectly designated as Section 3 in L171). Authors need to choose the best way, and the definition and meaning of seven genotypes has to be written firstly, with following results for analysis of these seven genotypes.
(17) L166, L169-170, and L240. Legends of Figures 1, 2 and 4 have to contain explanations of all figure panels separately: A, B, C, and D (for Figure 1), and A and B (for Figures 2 and 4).
(18) L192. ‘ANOVA’ is the abbreviation of method’s name and it has to be in capitalized letters.
(19) L198-208. In this fragment, Authors use one mathematical formula with explanation of the components, designated as ‘symbols’ by three small letters as follow: ‘g’ (genotype); ‘e’ (environment or growing season) and ‘r’ (repeat). It is better to use these designated symbols in Italics case, as it is present in the formula (L200). The second point is about attached two letters ‘th’ following immediately after the symbols: ‘gth’, ‘eth’, and ‘rth’. In my understanding, ‘th’ here is a suffix, which used to form ordinal or fractional order, and it is answering the question ‘which from several ones’? This is easy to explain with constituting numbers: ‘5 (five) – 5th (fifth)’; ‘10 (ten) – 10th (tenth)’. In this case, Authors have to use ‘th’ in superscript case in all occurrences of ‘gth’, ‘eth’, and ‘rth’. I cannot show it in the plain text but hopefully Authors can understand that symbols (‘g’, ‘e’ and ‘r’) have to be in Italics case while ‘th’ has to be in superscript non-Italics case.
(20) L258. Table 2. Inside of the Table, the column with the Head ‘df’ (Degrees of freedom) is useless because it indicates ‘1’ only everywhere. Therefore, Authors have to delete this column but insert a short statement, probably in the Footnote of the Table 2 (L260) that ‘Degrees of freedom (df) was = 1 for all presented calculations’.
(21) L333. The term ‘gene deletion’ is confusing and is used incorrectly because it can cause misunderstanding that ‘entire gene was deleted’, which is the wrong statement. I suppose Authors want to say as follow: ‘a few varieties had some deletions in Vrn and Ppd genes’.
(22) L349-350, L394-395, L400. The term ‘winteriness or semi-winteriness’ seems grammatically correct but this is very strange to see a combination like this: ‘winteriness or semi-winteriness wheat varieties’. There is simple and widely usual term ‘winter wheat varieties’. Why some ‘new and strange terms’ have to be introduced in this manuscript? This is unclear. Please correct for ‘winter wheat’. The second term ‘semi-winteriness’ is suspicious, and if it means ‘facultative’ (either winter or spring type of wheat plant development), please see Major point No 1. If Authors use the term ‘semi-winteriness’ with meanings as ‘early flowering (maturing) winter wheat’ only, please use this simple term.
(23) L390. The term ‘sunshine’ has to be replaced for more scientific and with clear meaning ‘long day-length in…’.
(24) L453-454. Please check the indicated reference (Dixon, 2009). This is unclear phrase about “the study on late flowering and early mature varieties”. What does this mean? Is this citation from Dixon, 2009 or the interpretation of Author’s of the manuscript? In any case, please check, clarify and add a phrase or a sentence with explanation why ‘LATE flowering and EARLY mature’ were mentioned? By the way, the reference for Dixon, 2009 in L528 is very strange. Even, it was indicated for CIMMYT, in current form, this is unacceptable format of the reference.
Author Response
Comments and Suggestions for Authors
The manuscript ‘Biology-1385222’ deals with very interesting and important topic of Time to anthesis (or flowering) trait in relation to genotypes and growing seasons in wheat. The manuscript is written in logical style, with good Introduction, appropriate M&M. important Results and interesting Discussion. Therefore, the manuscript can be conditionally accepted with a subject to minor revision. Authors have to address major comment and minor corrections indicated below carefully and properly to make their manuscript ready for the acceptance and publishing.
Major comment:
(1) Authors must clarify that they used only winter but not spring wheat in their study. This is particular clear in L125 and L217 that plants were grown from October over winter and harvested in June. Spring wheat cannot grow in such vegetation period. However, not only winter wheat but also ‘facultative’ wheat can be grown overwinter. Facultative wheat accessions are those that can be used either as winter or as spring wheat. Therefore, in fact, 100 studied wheat cultivars might represent only winter wheats or a mixture of both winter and facultative wheats. Authors must clarify the developmental type or types of used wheat germplasms. If all 100 studied wheat accessions were winter only, it will be relatively simple and minor corrections, ‘winter wheat’ must be inserted, at least, in the Title (‘…anthesis date of winter wheat sown…’), in Abstract, Keywords, at the first occurrences in Introduction and in the Discussion, and it has to be carefully described in M&M section.
However, the situation can be transformed into major changes, if at least one (or more) from 100 studied wheat accessions were facultative, and such wheat accessions had either winter or spring type of development. In this case, Authors must split and show their analyses separately, for winter and facultative wheats. This is not ‘a tragedy’ but it is very important to provide and publish correct and absolutely clear information, avoiding any potential misleading of Readers. If both winter and facultative wheats were used, the Title has to be modified, for example as follows: ‘Understanding effects of growing seasons, genotypes, and their interactions on the anthesis date of winter-growing wheat in North China’. In this case, ‘winter-growing’ means a mixture of both winter and facultative accessions of wheat. Following changes must be included in the text accordingly but additional analyses of two groups (winter and facultative wheats) and their comparison will be required.
Response: We agree with this concern. All 100 studied wheat accessions were winter-habit only. The varieties we selected are all adaptive historical varieties and currently sown varieties in Beijing. Besides, the present study site located at north edgy of region that allow winter wheat growth. Therefore, only winter-habit varieties can survive across winter. For this concern, we clarify the wheat germplasms are winter wheat across the entire manuscript.
Minor corrections/notes:
(2) L10-11, L69-77. Authors must be careful and indicate Latin names of plants species and names of genes in Italics: ‘Triticum aestivum’; ‘Vrn-1, Vrn-2, Vrn-3, and Vrn-4’ genes; and so on.
Response: We accept this concern and have made the corresponding revisions.
(3) L20, L24, L50, L244, L261, L321, L356, L384, L387, L460 and maybe in other parts of the manuscript. The term “wheat-anthesis date” is unclear, particular with using of a hyphen between words ‘wheat’ and ‘anthesis’. What does this mean and what Authors want to say? Starting from the Title and Abstract and through entire manuscript, Author also use correct term ‘anthesis date’, which is a synonym of ‘flowering date’, and it means ‘number of days until anthesis or flowering’. Authors have to correct all terms ‘wheat-anthesis date’ in the manuscript.
Response: We accept this concern and have made the corresponding revisions. We changed “wheat-anthesis date” to “anthesis date” across the entire manuscript.
(4) L34-35 and L328. The term ‘biological stress’ or ‘biological stressor’ is non-typical. It is better to change it for ‘biotic stress’. In all cases Authors have to use more popular terms ‘abiotic and biotic stresses’.
Response: We accept this concern and have made the corresponding revisions.
(5) L42. Mistyping ‘phenological’.
Response: Agreed. Yes, it’s mistyping. Thanks for correcting us.
(6) L43-44. Please re-phrase the sentence: “Crop breeding is a key mechanism for adaptation of cropping systems to local climate”. Crops breeding cannot be a mechanism… ‘Crop breeding’ is the ‘science’ or ‘scientific direction’, which studies (or based on) adaptation of cropping systems…
Response: We accept this concern and have made the corresponding revisions.
(7) L57-58. Authors used incorrect statement: “…the homoelogous gene of three types of Ppd-1, namely Ppd-A1, Ppd-B1, and Ppd-D1”. It has to be changed for clear meaning. The proposed variants are as follows ‘…three homoelogous genes, Ppd-A1, Ppd-B1, and Ppd-D1’ or ‘…Ppd-1 gene with copies in three homeologous chromosomes in A, B and D genomes’.
Response: We accept this concern and have made the corresponding revisions.
(8) L62. The term “short-day plants” is unclear, particular for wheat. It is better to change it for better meaning as follows: ‘wheat plants in short-day conditions’ or ‘plants grown in short-day environment’.
Response: We accept this concern and have made the corresponding revisions.
(9) L78-81. The sentence starting from: “A study on the vernalization genes…” with the reference to [Harris et al., 2017], is unclear. Please split it for two sentences and explain better, what is happened in wheat with dominant and recessive alleles of both Vrn-A1 and Vrn-B1 genes and with isolated Vrn-B1 for time to anthesis.
Response: We accept this concern and have made the corresponding revisions.
(10) L124-125. The following phrase is unclear: “…a semi-moist temperate continental monsoon climate”. What does ‘semi-moist temperate’ means? Is it cold or hot? Please clarify.
Response: We accept this concern and have made the corresponding revisions.
(11) L134. What was a density of wheat plants? The phrase “from 200 cm length and 25 cm spacing” does not help. Please provide information for both: ‘number of plants in one metre of row (or distance between plants in a row)’ and ‘average number of plants per m2 (or another density characteristic)’.
Response: We accept this concern and have made the corresponding revisions. In this study, the planting is sown according to rows, one row is a variety, not according to the experimental block, so only the introduction of sowing quantity per meter is added.
(12) L139. The phrase “recessive expression of photoperiod” represents a poor laboratory jargon. Both terms ‘recessive’ and ‘expression’ have very strong definitions in genetics, where ‘recessive’ and ‘dominant’ alleles only are used. Similarly, ‘expression’ is used only for ‘gene expression’ as activation of mRNA production during onthogenesis or in response of plants to external action. Please modify for clear meaning, what Authors want to say with this phrase.
Response: We accept this concern and have made the corresponding revisions.
(13) L143-145. Please cut a part of the second sentence making it shorter and more clear, combining two parts together. Original: “The vernalization and photoperiod alleles were identified. The alleles of Vrn-A1, Vrn-B1, Vrn-B3, and Ppd-B1 (Vrn-A1, Vrn-B1, Vrn-B3, and Ppd-B1b, respectively) were recessive”. Proposed: ‘Vrn-A1, Vrn-B1, Vrn-B3, and Ppd-B1b were identified as recessive alleles for vernalization and photoperiod, respectively’.
Response: We accept this concern and have made the corresponding revisions.
(14) L147. Authors must provide a full list of 100 used wheat accessions, for example in the Supplementary Materials. Moreover, this list has to include elementary information for each wheat cultivar: Name, Registration number, Origin (Originator), Year of release, Pedigree, One or two sentences of brief description of main characteristics for each wheat cultivar. If this is not acceptable for Authors, please insert here your statement as follows: ‘The list of 100 used wheat accessions represents the confidential information and cannot be released in open publicly-available papers due to … [insert your explaination, why the list cannot be provided]’. Only with the list of 100 used wheat accessions, it is possible (at least theoretically) to reproduce the presented results.
Response: We accept this concern and provide a full list of 100 used wheat accessions at the end of the article..
(15) L148-149. Please provide a reference for ‘Chinese Germplasm Resource Bank’. Web-site link would be very good but Authors have to be sure that web-site has English version for Readers.
Response: We accept this concern and have made the corresponding revisions.
(16) L151-164 and Figure 2A, and L179-185 and Table 1. Authors have to think and re-arrange the order of these two fragments with Figure 2A and Table 1. This is because results for seven genotypes (A1B1-A4B2) were shown and described earlier in sub-section 2.2 but their definition and meaning were provided later, in sub-section 2.3 (which was incorrectly designated as Section 3 in L171). Authors need to choose the best way, and the definition and meaning of seven genotypes has to be written firstly, with following results for analysis of these seven genotypes.
Response: We accept this concern and have made the corresponding revisions.
(17) L166, L169-170, and L240. Legends of Figures 1, 2 and 4 have to contain explanations of all figure panels separately: A, B, C, and D (for Figure 1), and A and B (for Figures 2 and 4).
Response: We accept this concern and have made the corresponding revisions.
(18) L192. ‘ANOVA’ is the abbreviation of method’s name and it has to be in capitalized letters.
Response: We accept this concern and have made the corresponding revisions.
(19) L198-208. In this fragment, Authors use one mathematical formula with explanation of the components, designated as ‘symbols’ by three small letters as follow: ‘g’ (genotype); ‘e’ (environment or growing season) and ‘r’ (repeat). It is better to use these designated symbols in Italics case, as it is present in the formula (L200). The second point is about attached two letters ‘th’ following immediately after the symbols: ‘gth’, ‘eth’, and ‘rth’. In my understanding, ‘th’ here is a suffix, which used to form ordinal or fractional order, and it is answering the question ‘which from several ones’? This is easy to explain with constituting numbers: ‘5 (five) – 5th (fifth)’; ‘10 (ten) – 10th (tenth)’. In this case, Authors have to use ‘th’ in superscript case in all occurrences of ‘gth’, ‘eth’, and ‘rth’. I cannot show it in the plain text but hopefully Authors can understand that symbols (‘g’, ‘e’ and ‘r’) have to be in Italics case while ‘th’ has to be in superscript non-Italics case.
Response: We accept this concern and have made the corresponding revisions.
(20) L258. Table 2. Inside of the Table, the column with the Head ‘df’ (Degrees of freedom) is useless because it indicates ‘1’ only everywhere. Therefore, Authors have to delete this column but insert a short statement, probably in the Footnote of the Table 2 (L260) that ‘Degrees of freedom (df) was = 1 for all presented calculations’.
Response: We accept this concern and have made the corresponding revisions.
(21) L333. The term ‘gene deletion’ is confusing and is used incorrectly because it can cause misunderstanding that ‘entire gene was deleted’, which is the wrong statement. I suppose Authors want to say as follow: ‘a few varieties had some deletions in Vrn and Ppd genes’.
Response: Agreed. We appreciate reviewer provide this statement.
(22) L349-350, L394-395, L400. The term ‘winteriness or semi-winteriness’ seems grammatically correct but this is very strange to see a combination like this: ‘winteriness or semi-winteriness wheat varieties’. There is simple and widely usual term ‘winter wheat varieties’. Why some ‘new and strange terms’ have to be introduced in this manuscript? This is unclear. Please correct for ‘winter wheat’. The second term ‘semi-winteriness’ is suspicious, and if it means ‘facultative’ (either winter or spring type of wheat plant development), please see Major point No 1. If Authors use the term ‘semi-winteriness’ with meanings as ‘early flowering (maturing) winter wheat’ only, please use this simple term.
Response: We accept this concern and have made the corresponding revisions.
(23) L390. The term ‘sunshine’ has to be replaced for more scientific and with clear meaning ‘long day-length in…’.
Response: We accept this concern and have made the corresponding revisions.
(24) L453-454. Please check the indicated reference (Dixon, 2009). This is unclear phrase about “the study on late flowering and early mature varieties”. What does this mean? Is this citation from Dixon, 2009 or the interpretation of Author’s of the manuscript? In any case, please check, clarify and add a phrase or a sentence with explanation why ‘LATE flowering and EARLY mature’ were mentioned? By the way, the reference for Dixon, 2009 in L528 is very strange. Even, it was indicated for CIMMYT, in current form, this is unacceptable format of the reference.
Response: We accept this concern and have made the corresponding revisions.
